# In Search of a Role for Extracellular Purine Enzymes in Bone Function

**DOI:** 10.3390/biom11050679

**Published:** 2021-04-30

**Authors:** Mariachiara Zuccarini, Patricia Giuliani, Francesco Caciagli, Renata Ciccarelli, Patrizia Di Iorio

**Affiliations:** 1Department of Medical, Oral and Biotechnological Sciences, University of Chieti-Pescara, Via dei Vestini 29, 66100 Chieti, Italy; mariachiara.zuccarini@unich.it (M.Z.); patricia.giuliani@unich.it (P.G.); patrizia.diiorio@unich.it (P.D.I.); 2Center for Advanced Studies and Technologies (CAST), University of Chieti-Pescara, Via L. Polacchi, 66100 Chieti, Italy; f.caciagli@unich.it; 3StemTeCh Group, Via L. Polacchi, 66100 Chieti, Italy

**Keywords:** bone homeostasis, extracellular purines, extracellular metabolizing enzymes

## Abstract

Bone is one of the major tissues that undergoes continuous remodeling throughout life, thus ensuring both organic body growth during development and protection of internal organs as well as repair of trauma during adulthood. Many endogenous substances contribute to bone homeostasis, including purines. Their role has increasingly emerged in recent decades as compounds which, by interacting with specific receptors, can help determine adequate responses of bone cells to physiological or pathological stimuli. Equally, it is recognized that the activity of purines is closely dependent on their interconversion or metabolic degradation ensured by a series of enzymes present at extracellular level as predominantly bound to the cell membrane or, also, as soluble isoforms. While the effects of purines mediated by their receptor interactions have sufficiently, even though not entirely, been characterized in many tissues including bone, those promoted by the extracellular enzymes providing for purine metabolism have not been. In this review, we will try to circumstantiate the presence and the role of these enzymes in bone to define their close relationship with purine activities in maintaining bone homeostasis in normal or pathological conditions.

## 1. Introduction

Bone is a highly mineralized structure constituting the body skeleton, which performs functions ranging from support to locomotion and posture to protection of the main internal organs and brain. Bone is also essential for hematopoiesis, ensured by cells present in the marrow inside bones, as well as for the storage of minerals, the function of which is regulated by factors produced within bone itself such as growth factors and hormones (i.e., bone morphogenetic proteins and calcitonin, respectively) [1,2].

Noteworthy, the bone undergoes continuous remodeling throughout life in physiological conditions, thus allowing a harmonious development of the body from birth to maturity as well as the maintenance of skeletal integrity and functionality throughout all stages of life. In healthy conditions, remodeling is the result of balanced deposition and resorption of the bone matrix and minerals operated by the coordinated activity of the main bone cells such as osteoblasts, osteoclasts, and osteocytes. This process is highly activated by bone trauma or infectious/inflammatory diseases to replace damaged tissue.

Osteoblasts are bone-forming cells deriving from differentiation of mesenchymal stem cells (MSCs) contained in the stroma. They are found arranged in arrays around the central sinus of the bone marrow and represent about 5% of total cells. During bone formation, osteoblasts secrete new organic bone matrix and support its mineralization. They also modulate osteoclastogenesis together with other cells (osteocytes and stromal cells).

Osteocytes deriving from the last stage of osteoblastic differentiation constitute 95% of bone cells. They are interconnected via dendritic processes, thus forming a tridimensional network within the mineralized bone matrix able to sense and to respond to mechanical forces. Moreover, they regulate osteoblast mineralization and appear to have an important crosstalk with MSCs to be recruited to maintain the bone function [3].

Finally, osteoclasts, deriving from the fusion of hematopoietic mononuclear progenitors, are giant multinucleated cells and represent less than 1% of bone cells. Osteoclasts are the only cells able to resorb bone tissue. For major information, please consult excellent and comprehensive reviews (i.e., [4,5]).

As mentioned above, different substances, many of which are produced within the bone itself, control its growth and remodeling [6]. Among these, purines may be included. These compounds, in addition to being widely present inside all types of cells, can act as extracellular signaling molecules capable of interacting with specific membrane receptors and producing a wide variety of effects in many cells and tissues. While this aspect has been extensively investigated in relation to bone homeostasis [7], less attention has been paid to the role of enzymes related to the metabolism of extracellular purines, which can profoundly influence the activities of these substances in the same context. In this review, we will examine the most recent literature, highlighting what is currently known about the role of extracellular purine degrading enzymes in bone functions, with the aim of attracting attention and stimulating research in this regard.

## 2. Brief Overview on the Activity of Purines and Related Receptors in Bone Function

Purines are endogenous compounds well known as fundamental cell constituents of nucleic acids together with pyrimidines; moreover, adenine- and guanine-based compounds participate in a number of intracellular biochemical reactions, acting as second messengers (cAMP and cGMP), source of energetic fuel to cells (mainly ATP) and modulators of the activity of G proteins coupled to the stimulation of metabotropic receptors (GTP-GDP) [8,9]. The transfer of ADP-ribose units from nicotinamide adenine dinucleotide (NAD^+^) to acceptor proteins, known as ADP-ribosylation, which causes post-translational modifications affecting numerous cellular events including bone biology, should also be mentioned [10]. Defects in the activity of the aforementioned intracellular compounds can obviously also alter bone morphology and functions. However, this is beyond the aim of this review, which focuses on the role of purines and related degrading enzymes acting at extracellular level.

Of note, these compounds, mainly nucleotides, are actively released from cells under basal conditions or in response to various nontoxic stimuli via different secretory mechanisms. Moreover, purines nucleotides and nucleosides can be lost from cells through membrane leakage lesions [11]. In any case, once in the extracellular fluid, purines are able to activate specific receptors, the cloning, expression, and activity of which have been investigated mainly in relation to adenine-based compounds [12], whereas less is known about guanine-based substances [13]. More in detail, purine receptors are grouped into two families named P1 and P2 receptors, which are responsive to adenosine (ADO) and mostly to ATP, respectively. The P1 family comprises four subtypes (A_1_, A_2A_, A_2B_, and A_3_), which are metabotropic receptors linked to different G proteins [14], whereas the P2 family is further subdivided into two subgroups, namely P2X and P2Y receptors. There are seven ionotropic P2X receptors (P2XR), which are activated by ATP, and eight metabotropic P2Y receptors (P2YR), of which P2Y_1_R respond to ATP and ADP, P2Y_2,4,6_ subtypes are mainly sensitive to uridine-based nucleotides, while P2Y_12, 13_ sites respond to ADP and P2Y_14_R are sensitive to UDP-glucose recently reviewed by [15]. The interaction of adenine nucleotides and nucleosides with the aforementioned receptors, which are mostly present in all body cells although with different levels of expression and activity, produces huge effects, contributing to the control of the functions of numerous tissues/organs [15,16], including bone [7].

As previously mentioned, bone undergoes continuous remodeling throughout life, which is especially evident in individuals exposed to particular conditions such as astronauts, who show bone loss, or in athletes submitted to intense exercise, in whom there is bone gain. Osteoblasts and osteoclasts are primarily responsible for this process and normally act in a concerted and balanced fashion to ensure proper bone growth and functioning. Purines contribute to the homeostasis of this system. Thus, before exposing the importance of purine degradation in this context, it is essential to mention how ATP is released within bone as well as the activity exerted by extracellular purines on the main functions of this structure.

First, mechanical forces are the main promoters of bone remodeling, to which an increased release of ATP contributes. About this, a recent paper by Inoue et al. [17] showed that upon a mechanical stress, osteoblasts deriving from rodent dental tissues release ATP by a vesicular nucleotide transporter (VNUT), the expression and function of which is regulated by different types of stimuli. An increase in ATP release from MC3T3-E1 osteoblasts was also observed as a consequence of mechanical signals (oscillatory fluid flow). Interestingly, it could be enhanced by the copresence of parathyroid hormone and was coupled to increased cell proliferation via P2Y_2_ receptors (P2Y_2_R) [18]. Additionally, Micholajewicz and coll. recently pointed out that mechanically stimulated ATP release from murine bone cells could be regulated by a reversible membrane disruption, which did not cause cell death but a rapid repair [19]. This event would be mediated by P2Y_14_ receptors (P2Y_14_R), the inhibition of which determined cell hypersensitivity to mechanical stimulation [20]. Interestingly, the same research group has developed a theoretical model to determine the amount of ATP released from cells and the propagation of the purinergic signal following a mechanical “injury” (that is, a rapidly reversible perturbation/damage of the membrane). Based on their calculations, the total amount of ATP released would depend on spatial factors, i.e., the maximum distance from the input/injury that stimulates purinergic responses. For example, a lesion with an initial size of about 100 nm (d_0_), which repaired within a half-time of ~15 s, was able to produce an ATP concentration of about 10 μM at the surface of the mechanically stimulated cells. This value was comparable to previous experimental measurements performed by the same researchers. Their mathematical model also confirmed that the maximum concentration of extracellular ATP is proportional to the severity of cell damage, so that activation of P2Y_1_R and P2Y_13_ receptors (P2Y_13_R), sensitive to ADP, can only occur as a result of large tissue lesions, in which the quantity and thereby the speed of ATP conversion to ADP is greater [21]. Therefore, any stimulus, depending on its initial intensity, would provoke a reaction in a more or less limited tissue area, involving P2 receptors with different affinity towards ATP or ADP.

Secondly, from a very general point of view, purine nucleotides, by activating various P2 receptors, modulate the differentiation process of bone cell precursors in a different way. Data obtained by administering exogenous compounds to osteoblasts, osteoclasts, or to their progenitors have shown that the P2 receptors mainly involved in the maintenance of bone homeostasis are high- and low-affinity ATP receptors such as P2Y_2_R or P2 × 1 (P2 × 1R) and P2 × 7 (P2 × 7R) receptors, respectively. Thus, while a compressive force intermittently applied to osteoblasts derived from human mandibular bone released ATP which, by activating P2 × 1R, promoted osteoblast differentiation [22], stimulation of P2 × 7R by the selective agonist 2’(3’)-O-(4-benzoylbenzoyl)adenosine-5’-triphosphate (BzATP) suppressed the differentiation of primary human osteoblasts [23]. However, we recently demonstrated in adipose-derived mesenchymal stromal cells (ADSCs) that only a prolonged stimulation of P2 × 7R inhibited the osteogenic differentiation of these cells, while endogenous ATP, physiologically interacting with the same receptors, sustained their differentiation [24]. Mechanical overloading applied to hematopoietic progenitor cells also promoted release of ATP, which, interacting with P2 × 7R, was able to promote the differentiation of osteoclast precursors [25]. As well, increased levels of extracellular ATP, interacting with P2 × 7R present on fibroblasts and osteoblasts, induced the generation of the receptor activator of nuclear factor kB ligand (RANKL), which further activated osteoclastic alveolar bone resorption and bone loss in periodontitis [26]. Very recently, it was found that also the P2Y_14_R activated by UDP-sugar reduced, in addition to the responsiveness to mechanical stimuli and the differentiation of osteoblasts, while stimulating their proliferation [20]. Likewise, low concentrations of ATP and UTP strongly inhibited osteogenesis in osteoblasts via P2Y_2_R. Interestingly, the P2Y_2_R performed their activity by primarily regulating extracellular ATP levels in both osteoclasts and osteoblasts [27].

In parallel, there are effects promoted by ADO, the receptors of which have been found in bone cells [28]. More in detail, all ADO receptors (AR) are present in the osteoblast precursors, that is the MSCs of both rat and human bone marrow, albeit with different levels of expression [29,30,31]. During their differentiation into osteoblasts, A_2B_ receptors (A_2B_R) are upregulated during the first days of this process, to be then downregulated [29], while A_1_ receptors (A_1_R) and A_2B_R prevail in mature osteoblasts [32]. Accordingly, osteoblast differentiation was greater when cells were exposed to the nonspecific ADO receptor agonist, NECA, an effect inhibited by the A_2B_R antagonist, MRS1706, but not by the A_2A_ receptor (A_2A_R) antagonist, SCH442416, while the A_2A_R agonist, CGS21680, had no effect on bone differentiation [29]. However, using pharmacologic inhibitors and A_2A_R knockout (KO) mice, the A_2A_R have been shown to promote the proliferation of bone marrow-derived MSCs [33]. Furthermore, A_2A_R downregulate during osteogenic differentiation so that their expression is reduced in differentiated osteoblasts, while the reverse occurs during chondrogenesis and is accompanied by a reduction in CD73 activity. This suggests that changes in A_2A_R/CD73 expression levels may direct cells towards osteogenic differentiation [34].

All ADO receptors are also expressed in osteoclasts [32] and there is evidence for a role of ADO in osteoclast maturation and activity. In particular, the A_1_R are constitutively activated and required for proper differentiation and function of osteoclasts, while A_2A_R activation exerts the opposite effect, resulting in the inhibition of osteoclast function. However, the results are not always consistent with those above reported and the discrepancy has mainly been ascribed to the adoption of different experimental models. Little is known about the involvement of the other two adenosine receptors, A_2B_R and A_3_ receptors (A_3_R), in osteoclast differentiation and bone resorption [33].

## 3. Enzymes and Other Mechanisms Generally Involved in the Turnover of Extracellular Purines

As it occurs at the intracellular level, extracellular purines undergo an efficient metabolism, which is mainly aimed at controlling the quantity of active purine compounds that interact with the respective receptors and therefore the duration and amplitude of the purinergic signals. As reported above for ATP [21], all purine metabolism should be proportional to the amount of purines present in the extracellular fluid, thus avoiding an excessive interference with normal cell functions and, also, a possible receptor downregulation [35,36]. Recalling that ATP is the main compound actively released from cells in physiological conditions, whereas pathological cell membrane alterations also allow ADO release, it is not surprising that the enzymes deputed to degrading the nucleotides in the extracellular fluid are numerous and groupable, as shown below.

Ecto-nucleotidases, also known as ectonucleoside triphosphate diphosphohydrolases (NTPDases), rapidly hydrolyze ATP and ADP to AMP. They include four membrane-bound enzymes: CD39, also known as NTPDase1, which hydrolyzes ATP and ADP with equal potency; CD39L1, known as NTPDase2, which preferentially hydrolyzes ADP over ATP; NTPDase3 and NTPDase8, which, in contrast, preferentially hydrolyze ATP over ADP [37].

Extracellular ATP can be directly degraded to AMP and pyrophosphate (PPi) by ectonucleotide pyrophosphatase/phosphodiesterase family member 1 (ENPP1) or to ADP by ENPP3 [38]. Interestingly, membrane-bound tissue nonspecific alkaline phosphatase (TNAP), which belongs to the alkaline phosphatase family and is mainly involved in mechanisms controlling normal skeletal mineralization and pathophysiological abnormalities, subsequently hydrolyzes PPi to inorganic phosphate [39]. TNAP is also able to covert nucleotides, mainly AMP, to ADO [40]. Finally, prostatic acid phosphatase (PAP), a member of the acid phosphatase superfamily, also catalyzes AMP hydrolysis [41].

The hydrolysis of ATP and ADP by pyrophosphatases can be counteracted by the activity of extracellular (membrane-bound or soluble) enzymes such as adenylate kinase isoenzyme 1 (AK1) and nucleoside diphosphokinase (NDPK). These enzymes contribute to the regeneration of extracellular ATP by catalyzing reversible phospho-transfer reactions [42,43].

Unlike purine nucleotides, the formation of ADO and its subsequent degradation is due to a few enzymes. Indeed, AMP, resulting from the extracellular ATP metabolism, is degraded to ADO by ecto-5′-ectonucleotidase also known as CD73 [44]. Moreover, ADO can also be generated from nicotinamide adenine dinucleotide (NAD^+^) through the coordinated actions of ADP-ribosyl cyclase 1 (CD38), ENPP1, and CD73 [45]. Subsequently, ADO present in the extracellular medium can be partially transformed into inosine by the ecto-adenosine deaminase (e-ADA) and inosine, in turn, is metabolized into hypoxanthine by the activity of purine nucleoside phosphorylase (PNP) [46,47,48]. Findings above reported for the classes of purine degrading enzymes are summarized in Figure 1.

To understand why the metabolism of ADO is limited, it is important to underline that this nucleoside, once formed in the extracellular environment, is actively regained inside cells by specific (equilibrative and concentrative) transporters [49]. It is noteworthy that ADO uptake as well as e-ADA activity are essential mechanisms that terminate ADO signaling [50]. It should also be noted that ADO metabolites such as inosine and hypoxanthine can be taken up into cells by the same carriers mentioned above or by others [51], which help remove compounds from the extracellular fluid that need to be recovered. Indeed, they are utilized intracellularly in the so called “purine salvage pathway” to reconstitute nucleosides and nucleotides, which are essential for the biological functions of cells [48]. A more detailed description of the properties and activities of all enzymes involved in ATP and ADO metabolism has recently been reported in [52,53].

Altogether, we have drawn a complex picture from which it is evident that the life of extracellular ATP and ADO is carefully timed by their active metabolism, to which it has to be added the contribution of carriers to eliminate ADO and other metabolites from the extracellular microenvironment. Given the importance of the entire network, the interplay among purines-enzymes-carriers is now called “purinome” [54].

## 4. Role of Enzymes Deputed to Purine Metabolism in Bone Function and Diseases

The first kinetic characterization of NTPDase1/CD39 in bone was performed in 2003, by Demenis et al. [55], who studied the activity of this enzyme in alkaline phosphatase-depleted rat osseous plate membranes, obtained 14 days after implantation of demineralized bone particles in the subcutaneous tissue of Wistar rats. Findings therein reported demonstrated the presence of the enzyme as involved in the calcification process, even though at that time its precise location on cell or matrix vesicle (MV) membranes was uncertain.

Later on, Noronha-Matos et al. [56] performed a study aimed at characterizing NTPDases in bone marrow MSCs from postmenopausal women. The experimental design was very complex, mainly investigating the role of P2Y_6_ receptors (P2Y_6_R) in the cell differentiation process. The data showed that uracil nucleotides are important stimulators of osteogenic cells differentiation, mainly through the activation of those receptors coupled with increasing intracellular Ca^2+^ concentration. It is noteworthy that the activity of endogenous uracil nucleotides was balanced through specific NTPDases, the levels of which were found to be higher in differentiated cells than in proliferating cells, thus likely determining whether osteoblast progenitors were driven towards proliferation or differentiation.

A more recent study on gingiva-derived MSCs reported that these cells express CD39 which can promote restorative osteogenesis in an experimental animal models of osteoporosis (ovariectomize mice) or autoimmune arthritis, by stimulating the Wnt/β-catenin signaling pathway [57] and by increasing the availability and receptor-mediated activity of ADO [58], respectively. Likewise, human ADSCs were able to limit the development of experimentally collagen-induced arthritis by inhibiting, via CD39 signals, the RANKL-induced genesis of murine or human osteoclasts. The authors suggested the involvement of regulatory T lymphocytes possibly recruited by ADO deriving from CD39/CD73 pathway [59].

In addition to the activity of purine nucleotides mediated by the interaction with their receptors, it is important to remind that extracellular ATP can also act as a source of phosphate for osteoblast mineralization, that is deposition of hydroxyapatite (HA) in specific areas of the extracellular matrix (ECM). This complex process is due to the balanced activity of factors promoting and inhibiting calcification [60]. Indeed, ECM mineralization is inhibited by extracellular inorganic pyrophosphate (PP_i_) production, caused by NPP1 activity which catabolizes extracellular ATP into PP_i_ and AMP [61,62,63]. Intracellular PP_i_ is also transported outside cells by the ankylosis protein (ANK) [64]. Unlike NPP1 activity, TNAP, anchored to the cell membranes of osteocytes and chondrocytes, plays an opposite role, reducing the extracellular PP_i_ concentration to ensure an adequate P_i_/PP_i_ ratio for normal bone mineralization [65]. The presence of NPP1 and TNAP was also revealed on the membrane of MVs released from osteogenic cells and the biochemical reactions catalyzed by these enzymes contribute to metabolize ATP, reciprocally controlling the generation of inorganic PPi and Pi, necessary for the apatite crystal formation [66,67]. In agreement with these findings, NPP1 (*Enpp1^−/−^*) or ANK (*ank/ank*) KO mice develop soft tissue calcification as a result of reduced production or transport of PPi [68], while mice deficient in TNAP function (*Akp2^−/−^*) showed rickets and osteomalacia due to a blockade of HA crystal formation caused by an increase in extracellular PP_i_ concentrations [69]. Of note, acidosis, which likely mimics an inflammatory condition, upregulated NPP1 expression along osteoblast differentiation, while that of TNAP as well as bone mineralization were reduced, while ATP release was unaffected [70,71]. Again, orthodontic forces have been reported to stimulate alveolar bone remodeling, leading to orthodontic movement of the teeth. *Enpp1* mutant mice exhibited reduced tooth movement and response to orthodontic forces compared to wild-type mice as well as altered osteoclast/odontoclast distribution. The authors suggested that NPP1 loss of function could directly affect cell function/recruitment, indirectly altering periodontal remodeling [72].

Another enzyme that plays an important role in bone homeostasis is CD73 ectonucleotidase which regulates extracellular ADO levels through AMP metabolism. Indeed, CD73 KO mice show impaired osteoblast differentiation and reduced bone formation with development of osteopenia [73], likely due to the lack of ADO formation and interaction with A_2B_R, as demonstrated by in vitro experiments. CD73 has also been found to be important for bone repair following injury in aged mice [74]. Again, the expression of CD73 as well as of CD39, controlling the extracellular levels of available ADO and ATP, respectively, was decreased in hematopoietic and nonhematopoietic bone marrow cells in a mouse model of postmenopausal bone loss (ovariectomized mice deficient in estrogen) [75]. As expected, an A_2B_R agonist reduced bone loss also in this in vivo model. However, there were some differences in CD73 and CD39 dysregulation caused by estrogen deficiency among cells. Indeed, the percentage of cells bearing CD39 increased in osteoblasts while decreasing in osteoclasts. In contrast, the percentage of CD73 expressing cells decreased in both osteoblasts and osteoclasts. These findings suggest that while CD39 may play a different role in osteoblasts and osteoclasts, CD73 activity is crucial in modulating extracellular ADO levels under estrogen signaling and, thereby, the nucleoside effect on the osteogenic differentiation of progenitor cells. Consistent with these observations, previous results showed that cells from CD39 KO mice did not show a decrease in osteogenic differentiation when extracellular ADO uptake was inhibited [76]. Therefore, estrogen signaling is important for maintaining the coexpression of CD73 and CD39 as well as the presence of adequate levels of extracellular ADO during osteoblast and osteoclast differentiation. This might also be true in humans [77]. A summary of the findings here included is reported in Table 1.

## 5. Role of Purine Enzymes in Bone Inflammatory Conditions

There are a number of diseases developing in high-grade systemic inflammation, such as rheumatoid arthritis (RA), or in conditions of low-grade systemic inflammation like those observed in obesity, related or not to type 2 diabetes mellitus, and chronic kidney disease (CKD), which all exhibit altered bone structure and/or function, in addition to other organic dysfunctions.

Looking more in depth to these inflammatory diseases, some of them, such as RA, are characterized by great production of numerous cytokines (interleukin-1, interleukin-6, tumor necrosis factor) in the synovium that, in turn, activate osteoclasts and mediate cartilage and bone destruction of the joints, with a systemic effect leading to generalized bone loss [86]. Besides osteoclasts, immune cells including macrophages, mastocytes, dendritic cells, monocytes, plasma cells and B, T helper 1 (Th1), and 17 (Th17) lymphocytes, actively participate in tissue resorption [87,88]. Osteoporosis as well as osteoarthritis are present in obesity, where the release of adipokines from bone marrow fat can alter bone trophism, even though the role of these substances is still controversial [89]. An increased risk of fracture has been reported mainly in obese patients with type 2 diabetes mellitus, in whom the levels of the same cytokines are enhanced [90]. Similarly, a mineral bone disorder is evident in CKD, a pathology in which high levels of cytokines, including transforming growth factor-beta in addition to the others mentioned above, are correlated with the disease progression [91].

It is noteworthy that an inflammatory environment is also present in “physiologically” occurring osteoporosis. In particular, proinflammatory cytokines are among the mediators of postmenopausal osteoporosis. As well, low-grade systemic inflammation is seen in aged conditions (nicknamed “inflamaging”), in which a decrease in bone density is frequently observed [92,93]. Thus, severe or mild chronic inflammation is a mechanism often associated to bone loss.

In addition to contributing directly to bony remodeling and repair, purines exhibit well known pro- or anti-inflammatory and immunomodulatory effects [94,95]. Thus, ATP, likely liberated from damaged and/or necrotic cells during inflammation, mostly acts as one of the damage-associated molecular patterns (DAMPs), contributing to trigger a pro-inflammatory response, although DAMPs may also act to promote tissue healing [94]. Obviously, dysfunctions in the metabolism of ATP could exacerbate its proinflammatory activity. This is the case of a genetically inheritable TNAP alteration, due to mutations in the *ALPL* gene leading to the rare disease hypophosphatasia (HPP) in humans. TNAP deficiency, distinguished in seven major clinical forms, may be coupled to bone marrow edema, myopathies, tendinitis as well as to increased development of periodontitis in some mildest form of HPP. The cause of inflammatory reactions in bones and muscles has been referred to the accumulation of PP_i_ leading to calcium pyrophosphate dehydrate accumulation in the tissues of HPP patients [78]. However, a decrease in ATP degradation cannot be ruled out among factors contributing to the manifestations of the disease, as TNAP is able, under normal conditions, to degrade ATP when its levels are high [40].

However, in inflammatory conditions, there may be an increase in the activity of purine degrading enzymes aimed at favoring the formation of ADO, which, unlike ATP, exhibits anti-inflammatory properties [96,97]. This has been found in lymphocytes from RA patients, in which increased extracellular NTPDase and reduced ADA activities have been reported [81]. Similar results were obtained using animals with complete Freund’s adjuvant-induced arthritis [82,83]. In agreement with these data, some studies have emphasized the role of e-ADA in inflammation, so that the serum activity of this enzyme is considered an inflammatory marker in diseases such as RA, based on the evidence that ADA, by causing a reduction in ADO levels, can stimulate the activity of immune cells, possibly also leading to bone dysfunctions [94]. However, bony dysplasia is present in severe combined immunodeficiency (SCI) sustained by ADA deficiency, a condition in which degradation of ADO to inosine is lacking at both extracellular and intracellular level. The overall symptoms of the disease are likely to be mostly due to a severe and complex failure of the purinergic system [84]. In contrast, no overt bone dysfunction has been reported in an autoinflammatory disorder due to selective deficiency of ADA2, which is the less abundant of the two known isoforms, ADA1 and ADA2, and predominantly expressed in human immune cells. This disorder is, indeed, characterized by vasculitis, hematological disease, and immunodeficiency [85].

In light of the above results, it is clear that data from the investigation on the activity and metabolism of purine should be always evaluated taking in due account the experimental model adopted for the study. Thus, ADO, likely produced by the enzymatic activity of CD73 in precursors of human osteoblast grown under normal conditions, would stimulate interleukin-6 formation and inhibit osteoprotegerin production by these cells, thereby improving osteoclastogenesis and bone resorption [98]. In this model, however, the immune component is missing and, in any case, there is no inflammation, so ADO would tend to favor the production of factors by the osteoblasts to recruit osteoclasts, thus balancing the processes involved in bone remodeling. According to these data, local in vivo delivery of AR agonists promotes bone regeneration [99]. As well, it has recently been documented that the pro-osteogenic activity of a three-dimensionally-printed bioactive ceramic scaffold is enhanced by dipyridamole, a well-known inhibitor of ADO reuptake, in rabbits undergone surgical procedure for segmental radius defect [100]. Conversely, the lack of ADO signaling in CD73 KO mice allows the development of spontaneous arthritis associated with inflammatory symptoms [79]. Inflammation is also present in humans with CD73 deficiency; however, this syndrome is coupled, inter alia, with vascular calcification, arteriomegaly, and tortuosity as well as calcification in small joints, while no specific bone alterations have been reported [44].

## 6. Discussion

From literature examined and cited here, it is evident that purines, acting as signal molecules, are involved in continuous bone remodeling and repair, which is achieved with a balance between the pro-osteogenic and proresorbing activities of osteoblasts and osteoclasts. In this regard, purine nucleotides seem to play an important role on the control of osteogenesis, favoring the activity of osteoclasts and at the same time being the main source of phosphate to constitute the inorganic part of the matrix. In contrast, ADO appears to promote MSC osteogenic differentiation, primarily via A_2B_R, and help modulate inflammatory reactions via A_2A_R, when needed.

Noteworthy, the activity of purine compounds is finely tuned by their metabolism which occurs through a series of steps that lead to the released ATP being degraded to ADO, which in turn, can be converted into inosine and other metabolites or regained inside the cells (also together with its metabolites). Therefore, purine metabolism likely contributes to the overall process of bone homeostasis, limiting the presence of both ATP and ADO and, consequently, the overstimulation of the respective purine receptors.

The importance of these enzymes in bone homeostasis is witnessed by pathologies that emerge when there is a defect in their expression. Thus, inherited lack of TNAP or ADA in humans has been associated to bone alterations [78,85]. However, this is not always evident in other purine enzyme deficiencies in humans, such as CD73 deficiency, which has not been correlated with bone changes [44], while CD73 KO animals showed osteopenia or osteoporosis as well as spontaneous development of arthritis [79]. In this regard, it has recently been emphasized that this experimental model is not entirely adequate since there are important phenotypic differences between CD73-deficient mice and humans [80]. Therefore, more studies in humans are needed and research is now moving in this direction. In fact, a very recent survey in Danish children and adolescents has shown that some markers, closely related to bone turnover and including TNAP (referred to as bone alkaline phosphatase), vary with age and sex and knowledge of their levels may be useful for determining any early bone disorders [101].

In this context, an often overlooked aspect should be emphasized, when it comes to bone dysfunctions, namely the crucial importance of an efficient blood supply to the whole body thanks to an optimal circulation. As recently reported, the accumulation of calcium-phosphate crystals (i.e., HA) in the arterial wall as well as in cardiac valves has a significant impact on morbidity and mortality in several diseases such as CKD, osteoporosis, and diabetes patients by provoking severe cardiovascular events [102]. Interestingly, these pathological conditions are the same in which bone loss is also present. This can be explained by considering that there are some similarities between bone formation and pathological vessel calcification as well as in the contribution of purines and their metabolic enzymes to both processes. Indeed, the enzymes NPPs, CD39, CD73, and TNAP regulate the production and breakdown of the calcification inhibitor, PP_i_, and the calcification stimulator, P_i_, from extracellular nucleotides. This contribution, when balanced, is obviously beneficial for building/repairing bone, while it becomes detrimental to bone and also to vessels or cardiac valves and, thereby, to all the organs of the body, when there is an alteration in the purine enzyme function [103]. In particular, TNAP has very recently been indicated as a therapeutic target for cardiovascular calcification [104]. In our opinion, this aspect deserves further attention.

Finally, one of the most promising fields of investigation onto the role of purine enzymes in bone could be the inflammatory conditions of this tissue. The study of the levels of expression and/or activity of these enzymes in acute or chronic bone diseases could be important to understand if and how their modifications contribute to the resolution or chronicization of the inflammatory process and, therefore, to bone healing or damage worsening.

## 7. Conclusions

In conclusion, while there is a great deal of research on the contribution of purinergic compounds to ossification and bone repair processes, so far, no equivalent attention has been paid to the involvement of the enzymes that regulate the availability of these compounds at extracellular level. Yet they act like a balance needle in favoring or not the activity of the aforementioned compounds in bone remodeling and healing.

The role of purines and their metabolizing enzymes is gaining a great interest in oncology, where they are actively investigated as cancer prognostic markers and/or druggable targets [105,106,107,108,109]. The same happens in inflammation, where the role of purines and their degrading enzymes is recognized as important for a better pathogenetic understanding of acute and chronic inflammatory [92,93] or autoimmune diseases [110]. Interestingly, all these processes are characterized by a huge number of extracellular purines released from healthy cells or lost from damaged ones. The same should occur in bone diseases, where inflammation is present. Therefore, we do believe that also in this case the role of purine metabolizing enzymes closely related to the activity of purine compounds needs to be investigated in more detail.

We hope that this review will foster a new interest of researchers in this regard, which can hopefully lead to the discovery of new targets in the therapy of bone dysfunctions.

## Figures and Tables

**Figure 1 biomolecules-11-00679-f001:**
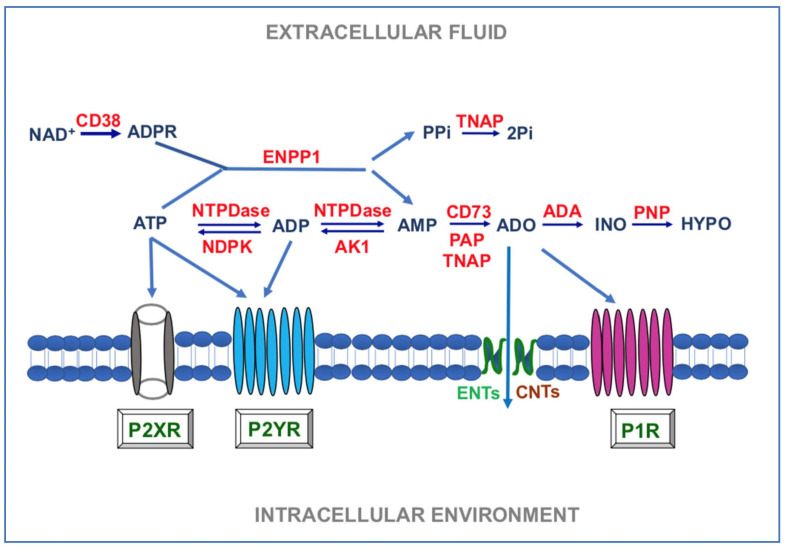
Main enzymes involved in the metabolism of extracellular purine. ADO: adenosine; ADA: ecto-adenosine deaminase; AK1: adenylate kinase 1; ADPR: ADP-ribose; CD38: ADP-ribosyl cyclase 1; CD73: ecto-5′-ectonucleotidase; CNTs: concentrative nucleoside transporters; ENPP: ectonucleotide pyrophosphatase/phosphodiesterase; ENTs: equilibrative nucleoside transporters; HYPO: hypoxanthine; INO: inosine; NTPDase: ectonucleoside triphosphate diphosphohydrolase (CD39); PAP: prostatic acid phosphatase; PNP: purine nucleoside phosphorylase; P1R, P2XR, P2YR: P1, P2X, P2Y receptors; PP1: pyrophosphate; Pi: inorganic phosphate; TNAP: tissue nonspecific alkaline phosphatase.

**Table 1 biomolecules-11-00679-t001:** Role of extracellular purine degrading enzymes in bone cells.

Type of Enzyme	Experimental Model	Activity in Bone	References
**CD39**	Alkaline phosphatase-depleted rat osseous plate membranes	Involvement in calcification process	[55]
	Human bone marrow MSC (postmenopausal women)	Together with NTPDases 2 and 3 regulation of osteoblast proliferation or differentiation caused by UDP activity on P2Y_6_R	[56]
	Mouse gingiva MSCs	Promotion of ADO availability and its reparative osteogenesis in ovariectomized or arthritic mice	[57,58]
	Human ADSCs	Inhibition of RANKL-induced genesis of mice/human osteoclasts	[59]
**NPP1**	Different mammalian tissues	Inhibition of ECM mineralization	[61,62,63]
	NPP1 (*Enpp1^−/−^*) KO mice	Development of soft tissue calcification as the result of reduced production or transport of PP_i_	[68]
	Rat osteoblasts	Acidosis/inflammation upregulate its expression.	[71]
	Rat osteoblasts/osteoclasts	NPP1 loss of function might directly affect cell function/recruitment, indirectly altering periodontal remodeling	[72]
**TNAP**	Mouse osteoblast-derived MVs	Assures an adequate P_i_/PP_i_ ratio for normal bone mineralization.	[65]
	Mice deficient in TNAP function (*Akp2^−/−^*)	Induction of rickets and osteomalacia due to blockade of HA crystals formation caused by an increase in extracellular PP1 concentrations	[69]
	Rat osteoblasts	Acidosis/inflammation downregulate its expression together with bone mineralization.	[70,71]
	Human deficiency	Various bone disorders	[78]
**CD73**	Human CD73 deficiency	Vascular calcification, arterio-megaly and tortuosity as well as calcification in small joints, without specific bone alterations	[44]
	CD73 KO mice	Impaired osteoblast differentiation and decreased bone formation with development of osteopenia	[73]
	Osteoblasts and osteoclasts	Its expression and activity together with that of CD39 controlled by estrogens in mice and man.	[75,77]
	CD73 KO mice	Development of spontaneous arthritis and inflammatory symptoms	[79]
	Human and mouse tissues	Differences in enzyme activity	[80]
**ADA**	Mouse model of RA	Decreased activity in arthritic mouse to preserve ADO levels and its anti-inflammatory and immuno-suppressive properties	[81]
	Animal model of induced arthritis	Activity altered by vitamin D3 given as an alternative treatment for chronic arthritis	[82]
	Arthritic rats	Activity attenuated by Quercetin together that of IFN-gamma and IL-4	[83]
	Human SCI	Bony dysplasia	[84]
	Human selective ADA2 defi-ciency	No reported bone alterations; vasculitis, hematological disease and immuno-deficiency were present	[85]

Legend: ADA, adenosine deaminase; ADO, adenosine; ADSCs, adipose-derived mesenchymal stromal cells; CD39, NTPDase 1; CD73, 5′-ectonucleotidase; KO, knockout; IFN-gamma, interferon gamma; IL, interleukin; MSCs, mesenchymal stromal/stem cells; MVs, matrix vesicles; NPP, ectonucleotide pyrophosphatase/phosphodiesterase; NTPDase, ectonucleoside triphosphate diphosphohydrolase; Pi, inorganic phosphate; PPi, pyrophosphate; RA, rheumatoid arthritis; SCI, severe combined immunodeficiency; TNAP, tissue nonspecific alkaline phosphatase.

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
