# Peer review of "In Search of a Role for Extracellular Purine Enzymes in Bone Function"

_biomolecules, 2021, doi:10.3390/biom11050679_

Round 1
Reviewer 1 Report
This is a nicely written review which covers current knowledge regarding the role of ecto-nucleotidases in bone. As the authors acknowledge this is an area which warrants further study. It discusses a broad topic well and contains reference to a wide range of published literature. It only needs some minor changes before it is suitable for publication:
- I think the review would benefit from a schematic diagram showing the key metabolism pathways.
- It would be good if there was more discussion of the highly localised effects of purine metabolism and as result purinergic signalling i.e. the rate of metabolism of nucleotides in one area of bone could be very different to another region. This means any functional effects could be limited to a very small area.
-
Some abbreviations are used in text without first being defined e.g. in section 2 ADO is not first referred to as adenosine.
- When referring to the P2Y receptors, the correct nomenclature is that the number should be subscript e.g P2Y1
Author Response
Please, see the attachment

Reviewer 2 Report
The paper of Zuccarini et al reviews the evidence for a role in bone function of extracellular purine enzymes, purinergic agonists and receptors. It lists a considerable amount of evidence relating to both physiological and pathological functions. Two Tables summarize purine compound degrading enzymes and their functional role, respectively. This is followed by a discussion section which emphasizes the role of purine compounds as signal molecules in bone remodeling and repair.
It is certainly meritorious to list and interpret all the partially diverse evidence on this topic.
The manuscript requires support by a language editor. Otherwise, I have no major queries.
Minor points:
p.2, l.4 up: ADP is an agonist also of P2Y1 receptors
In contrast to P2X receptors the number is subscript in P2Y receptors. In case of the adenosine receptors only the A is in capital letters.
- Table 1: ENPPs: Degrade ATP to AMP an PPi; TNAP: converts ATP directly to ADO. (otherwise, you need to exchange ADO for nucleoside)
Author Response
Please, see the attachment

Round 2
Reviewer 2 Report
The paper has been improved but concerns about style remain.
Before submitting, the term moderating needs to be exchanged for the correct term modulating (l. 7 Discussion section).
Author Response
We thank this Reviewer very much for his/her patience. Driven by his/her suggestions, we not only replaced the verb "moderate" with "modulate", as requested (see at pag. 10, last line of the new version of the manuscript, but we also took the opportunity to revise the entire manuscript, trying to improve its style.
We wish that the re-revised manuscript is now more understandable and easier to read.